# Transcriptional Regulation and Implications for Controlling *Hox* Gene Expression

**DOI:** 10.3390/jdb10010004

**Published:** 2022-01-10

**Authors:** Zainab Afzal, Robb Krumlauf

**Affiliations:** 1Stowers Institute for Medical Research, 1000 E. 50th, Kansas City, MO 64110, USA; zaf@stowers.org; 2Department of Anatomy and Cell Biology, Kansas University Medical Center, Kansas City, MO 66160, USA

**Keywords:** *Hox* genes, transcriptional regulation, transcription factors, nascent transcripts, gene regulation, coordinate regulation, enhancers

## Abstract

*Hox* genes play key roles in axial patterning and regulating the regional identity of cells and tissues in a wide variety of animals from invertebrates to vertebrates. Nested domains of *Hox* expression generate a combinatorial code that provides a molecular framework for specifying the properties of tissues along the A–P axis. Hence, it is important to understand the regulatory mechanisms that coordinately control the precise patterns of the transcription of clustered *Hox* genes required for their roles in development. New insights are emerging about the dynamics and molecular mechanisms governing transcriptional regulation, and there is interest in understanding how these may play a role in contributing to the regulation of the expression of the clustered *Hox* genes. In this review, we summarize some of the recent findings, ideas and emerging mechanisms underlying the regulation of transcription in general and consider how they may be relevant to understanding the transcriptional regulation of *Hox* genes.

## 1. Introduction

Animals display remarkable variety in their body plans and there is great interest in understanding the degree to which conserved and distinct mechanisms underlie this diversity in the formation and elaboration of basic body plans in animal evolution. In chordate evolution, there is emerging evidence for a deeply conserved regulatory network, involving transcription factors (TFs) and signaling pathways, that governs patterning along the anterior–posterior (A–P) body axis [1,2,3,4,5,6]. Remarkably, despite very different morphologies among chordates, many key TFs and components of major signaling pathways (e.g., Wnts and FGFs), known to regulate developmental processes, have been shown to be similarly aligned along the A–P axis. This suggests that regulatory interactions between signaling pathways and core TFs set up a conserved gene regulatory network (GRN) that guides the formation of the basic body plan and patterning of the A–P axis. However, the question of how TFs are coupled to these ancient signaling pathways and how they integrate responses to signaling gradients is not fully understood.

The highly conserved HOX family of TFs are an example of TFs that are coupled to this ancient GRN. *Hox* genes are known to play key roles in axial patterning and regulating the regional identity of cells and tissues in a wide variety of animals from invertebrates to vertebrates [7,8,9,10,11]. The clustered *Hox* genes exhibit an interesting property known as collinearity [12,13,14,15,16]. Genes in the four mammalian *Hox* clusters are all transcribed in the same 5′ to 3′ direction with respect to transcription, and the order of *Hox* genes in each cluster on a chromosome corelates with their temporal and spatial expression domains and functions along the A–P axis of developing embryos (Figure 1). These nested domains of expression generate a combinatorial *Hox* code, which provides a molecular framework that serves as a key regulatory step in specifying regional identities and properties of tissues along the A–P axis. A wide variety of studies in different species and cell culture models have revealed that the nested domains of *Hox* expression along the A–P axis arise in part through the ability of *Hox* clusters to integrate and respond to opposing signaling gradients, such as those of Retinoic acid (RA), Fibroblast growth factors (Fgfs) and Wingless related integration sites (WNTs) [5,17,18,19,20,21,22,23,24,25,26,27,28,29]. Hence, it is important to understand the regulatory mechanisms through which signaling pathways are able to coordinately control the precise patterns of the transcription of the clustered *Hox* genes required for their roles in specifying diverse morphologic features along the A–P axis.

In the case of RA signaling, *Hox* genes are direct transcriptional targets of retinoids, and their response to RA signaling involves retinoic acid response elements (RAREs) embedded within and adjacent to the *Hox* clusters [18,30,31,32]. These RAREs are *cis*-regulatory components of RA-dependent enhancers that provide regulatory inputs both locally on adjacent *Hox* genes and over a long range to coordinately regulate multiple genes in a *Hox* cluster [33,34,35,36,37,38]. This tightly clustered organization of *cis*-regulatory elements and the *Hox* genes they control raises interesting questions with respect to roles for chromosome topology, epigenetic modifications, dynamics of transcription and the underlying transcriptional mechanisms for how enhancers display selectivity or competition between genes, and they may be shared by multiple genes in a cluster [39]. It is important that these diverse aspects of transcriptional regulation are properly coordinated to ensure the right spatial and temporal patterns and appropriate levels of expression needed for their roles in axial patterning.

The advent of new technologies for investigating the dynamics of interactions that underlie the activation of transcription are generating surprising findings. These observations challenge the widely postulated role of stable long-term enhancer promoter interactions and the notion of a single RNA polymerase with a small number of components regulating transcription [40,41,42,43,44,45,46,47,48,49]. New models suggest that dynamic condensates and mechanisms involving a series of rapid and complex interactions underlie the activation of transcription and the regulation of gene expression. It will be interesting and important to understand how this newly emerging picture of the dynamic molecular mechanisms governing transcription plays a role in modulating the inputs controlling the coordinated expression of the clustered *Hox* genes. In this review, we summarize some of the recent findings, ideas and emerging mechanisms underlying the regulation of transcription in general and consider how they are relevant to the transcriptional regulation of *Hox* genes.

## 2. Regulatory Features

### 2.1. Enhancers

Enhancers were first discovered in simian virus 40 (SV40), where it was found that they function in an orientation-independent manner to stimulate transcription on heterologous genes [50]. Since then, a variety of analyses have revealed that animal genomes contain a large number of putative enhancers, out numbering coding genes [51,52]. It is challenging to identify *cis*-regulatory elements, such as enhancers, encoded in the genome through sequence analyses and computational methods alone [53]. Major efforts have been made to find ways of identifying and characterizing enhancers and their properties on a genome-wide and individual basis, which is important to facilitate our ability to decode regulatory information embedded in the genome [54,55,56,57,58,59,60]. While many development specific enhancers, including some of those discovered in the *Hox* clusters, are evolutionarily conserved [35,61,62,63], many adult or tissue-specific enhancers can be highly variable across species [64,65]. Even when enhancers are highly conserved, it can be challenging to understand the information content and the critical arrangements of the *cis*-elements that govern their ability to regulate expression [54,59,60]. Furthermore, highly conserved patterns of gene expression can arise through enhancers that display divergence [65]. Enhancers serve to stimulate transcription by integrating a variety of different regulatory inputs and binding sites for TFs to confer precise temporal, spatial and cell-type specific gene expression programs. Precise regulatory outputs from enhancers do not require that upstream factors have highly restricted domains of expression and can arise through the cumulative integration of weak, imprecise or wide-spread inputs by TFs [66,67]. The convergence of inputs can result in the integration of disparate and very broad patterns of regulatory signals into robust and tightly controlled specific outputs. Similarly, clusters of weak enhancers can synergize to serve as super enhancers to robustly regulate gene expression [68].

Enhancers can be located directly upstream of a gene or up to over a megabase away from its target gene promoter [69,70]. They frequently reside within introns of genes, even in ones they do not regulate, [70], and there is evidence for enhancers and *cis*-regulatory elements embedded in coding exons, including those of *Hox* genes [71,72,73]. Studies have shown that enhancer regions are themselves transcriptionally active. Several groups have demonstrated that non-coding enhancer RNAs are more than just transcriptional noise or byproducts of the transcriptional machinery, but are useful indicators in predicting active enhancers [74].

A challenge in identifying the targets of enhancer activity is that they can function independently of their orientation with respect to target genes and can make long-range enhancer–promoter contacts to more than the near adjacent genes [75]. Typically, an average vertebrate enhancer can be ~5 to 50 kb away from target promoters and ~1 to 10 kb away in the more compact *Drosophila* genome [39]. Intriguingly, the proximity of an enhancer to its target promoter required for functional activity is variable. Hence, there are no clear rules on how close enhancers should be positioned relative to the promoters they activate. Some studies have shown through proximity-dependent ligation techniques, such as 3C (Chromatin conformation capture), that enhancers physically come into contact with promoters, resulting in the activation of gene transcription [76]. Imaging approaches have shown that following the activation of genes, the distance between the enhancers and their target promoter tends to increase, suggesting a change in their interactions dependent upon their activity state [77]. This raises fundamental questions, such as, how do enhancers locate and distinguish between the target genes they activate; what confers enhancer–promoter specificity; and what degree of proximity is essential for the enhancer interactions required for gene regulation [39,55,69,75,78,79,80]?

These questions are relevant to understanding the regulation of the *Hox* clusters because of the high gene density and compact nature of the clusters. The enhancers embedded within and flanking an individual *Hox* cluster can display selective preferences, competition between genes and can regulate both near adjacent genes or act more globally on other genes in the complex. For example, in the mouse *Hoxb* complex, there are three RAREs in the middle of the cluster, two upstream and one downstream of *Hoxb4* (Figure 2A), which participate in mediating its response to RA by regulating multiple coding and long non-coding (lncRNAs) transcripts [33,35,37,81]. One of these RAREs (*DE-RARE*) is an essential *cis* element of an RA-dependent enhancer, which undergoes epigenetic modifications, and is required to coordinate the global regulation of *Hoxb* genes in hematopoietic stems cells [34]. This functional role for an enhancer raises many questions regarding the mechanisms through which the *DE-RARE* participates in regulating so many transcripts, how targets are selected and the dynamics of the process. Why, in contrast, do other enhancers embedded in the *Hoxb* cluster only appear to work on a single near adjacent gene [37,62,82,83,84]?

### 2.2. Transcription Factors and Their Impact on Enhancer Activity

One reason it is hard to identify and predict the functions of enhancers is related to our level of understanding of TFs and their properties [55]. TFs do not bind in isolation, and their binding specificities or other properties will depend on cofactors, interacting proteins and epigenetic states [85,86]. The DNA binding domains of the families of TFs generally recognize 6–12 base pair consensus motifs identified in vitro, and these TFs can activate and/or repress transcription [87,88,89,90]. However, it can be experimentally challenging to characterize the rules and basis for the context-specific activities of TFs in vivo. The *Hox* genes encode TFs that are thought to play highly conserved roles across species and the studies on their properties serve to illustrate some of the complex issues for understanding how TFs contribute to enhancer activity and function.

In vitro HOX proteins display very similar DNA binding properties [87,88,89,91,92,93,94,95]. However, their in vivo binding properties and functions can be distinctly different [96,97]. Some insights into the basis for these differences have come from experimental studies that have shown that the *Hox* genes have many auto- and cross-regulatory inputs in controlling their expression. The characterization of endogenous HOX-response elements has enabled insights into their in vivo properties and interactions with co-factors [38,83,85,98,99,100]. The binding specificity of HOX proteins arises not only from the homeodomain’s ability to recognize a short AT-rich motif (ATTA) defined in vitro, but in concert with other domains adjacent to the homeodomain (hexapeptide) and in different regions of HOX proteins, which can impact its specificity and ability to bind DNA [85,97,101,102,103]. The nature of this altered specificity for HOX proteins on target sequences is, to a large degree, related to their ability to interact with cofactors, in particular with members of the TALE family, PBX and MEIS (Figure 2B) [85,86,97,99,103,104,105,106,107]. HOX proteins interact with PBX proteins through multiple domains, which alters their target site preferences. HOX-PBX dimers bind to a bipartite HOX-PBC motif and can form a ternary complex with MEIS proteins [108,109,110]. This bipartite HOX-PBC motif is highly enriched in genome-wide analyses of downstream HOX target regions identified using chromatin immune precipitation (ChIP) assays [96,97,99,111,112,113], underscoring the important role TALE proteins play as cofactors in potentiating HOX functions.

While the clustered *Hox* gene family and its organizational features are highly conserved in animals, what about the functional roles of HOX proteins? Genetic analyses of *Hox* paralogous group members reveal many overlapping roles [114,115,116,117,118], but there is also evidence for specific transcriptional functions in vivo [97,119] and specific binding preferences that vary with cell-type specific [67,95]. Furthermore, it is unknown whether there is a conserved ancestral role for the duplicated and diverged paralogous *Hox* genes in vertebrates or if have they evolved new functions. Recent cross-species functional studies on the mouse HOX1 paralogs, which are related to *labial* in *Drosophila,* have revealed that despite an only 35% amino acid identity between the proteins, mouse HOXA1 has retained the ancestral activities of Labial [97]. The other mouse HOX1 paralogs, HOXB1 and HOXD1, display altered DNA binding properties and they have diversified to take on new functional roles. The basis for the retention of ancestral activity versus new functional properties maps to a small number of subtle amino acid changes in domains outside of the homeodomain that appear to alter interactions with TALE proteins [97]. Contributions of small, previously uncharacterized domains in other HOX proteins have also been found to alter binding properties though interactions with TALE cofactors [102], indicating that this may be a general mechanism used in evolution for modulating the activities and interactions of HOX proteins and other TFs. This illustrates the challenges in predicting the binding properties and functional roles of closely related TF proteins and may help to explain how they have diverse impacts on the enhancer activity.

Recently, HOX proteins bound to specific enhancers have been shown to confer tissue-specific transcriptional outputs by altering the activity of TALE proteins [120]. There is evidence that HOX proteins have pioneer activity, such that the HOX TFs of posterior *Hox* genes have different affinities for compacted chromatin [121] or can alter the accessibility to chromatin [90]. Therefore, for a deep level of understanding of how *Hox*-responsive enhancers work, it is important to understand how many and what kinds of TFs bind to an enhancer, the rules or syntax of their spacing and organization, the positions of nearby motifs and the identity of motifs that can serve to synergize or antagonize their activities [54]. Furthermore, the modifications of epigenetic states can alter these relationships to aid or prevent binding and activity. For a mechanistic insight into the coordinate regulation of *Hox* genes, we need to understand the enhancer “grammar” that dictates TFs binding to the DNA and their correlation with functional outputs.

### 2.3. Genome Organization and Its Impact on Enhancer Activity

Chromatin is spatially organized in the nucleus and several studies have shown that distinct organized chromatin domains facilitate processes such as the regulation of gene expression, DNA replication and repair and chromosome compaction [122]. The mechanisms underlying how the dynamics of chromosome arrangements impact function is still unclear [123]. Technological advances, especially chromatin capture assays (e.g., Hi-C), have shown that genomes are organized into Topologically Associated Domains (TADs) [124,125,126]. TADs physically confine segments of the chromatin, and there is evidence suggesting that enhancers and their target genes are often within the same TAD [39,127]. Enhancers can interact with other enhancers in a TAD and generally do not appear to interact with and activate genes outside the TAD [124,125,126]. With respect to *Hox* genes, analyses have shown that long range enhancer–promoter contacts are restricted within the TAD containing the *Hoxa* cluster [128]. Interestingly, TAD boundaries are not fixed and can vary. Elegant genetic and molecular studies from the Duboule group have shown that TAD boundaries in the *Hoxd* cluster can shift over developmental time, altering the ability of a long-range limb enhancer to regulate expression in embryos and digits [124,129,130,131,132]. Enhancer promoter interactions within a TAD appear to correlate with gene expression levels, and enhancer interactions outside of a TAD may be inhibited by the boundary elements present on the edges of TAD [131,133,134]. These boundary elements contain insulator proteins such as CTCF, and this TAD-based organization of chromatin is thought to help prevent unwanted enhancer interactions and effects on gene expression [133]. The importance of CTCF sites in regulating genes in the *Hoxa* cluster have been shown in cells and embryos [128,131,135]. Hence, the disruption of TAD boundaries can be associated with diseased states [127,136,137,138].

The regulation of genes by *cis* elements within TADs implies that there must be features and mechanisms that influence stable and/or dynamic contacts among the regulatory elements and their target promoters to facilitate precise transcriptional activity. Models, such as linking [139] and tracking [140], have been proposed for how enhancers can activate genes, but new data indicating very rapid dynamics make it challenging to explain or predict enhancer activity with these existing models [39]. A mechanism that currently seems plausible is based on the presence of chromatin loops that can be formed using a loop-extrusion model [141,142]. The loop-extrusion model postulates that DNA is squeezed out into a loop aided by Structural Maintenance of Chromosome (SMC) proteins, such as cohesin and condensin. They clasp DNA and these proteins stop moving DNA through the loop when they reach properly oriented CTCF sites [122,143]. This mechanism also allows for loops to form within the TADs, and dynamic enhancer contacts can thus be envisioned as the formation of different loop conformations to activate or inhibit transcription. It has been shown for *Hoxb* genes that active genes loop out of chromosome territories, and this looping out corelates with transcriptional activity [144]. The whole *Hoxb* cluster is in one TAD [124], and it can serve as an example to postulate different loop confirmations associated with previously identified enhancers involved in activating specific *Hoxb* genes (Figure 3A) [18,33,34].

TADs have been shown to enable regulatory contacts that take place within their boundaries [145], and therefore appear critical for the maintenance of gene regulation. Intriguingly, however, mutations in CTCF binding sites [125,146] and of cohesion [147], do not appear to disrupt genome compartmentalization or significantly impact gene expression even though they disrupt TADs [148]. This indicates that there is not an absolute correlation between TAD formation and the proper regulation of gene expression. It could be that the formation of a TAD does not necessarily restrict or promote all the enhancer contacts, it biases them [149]. Indeed, for early developmental enhancers, it has been observed that they may already be near their target promoters even before TADs are formed and the respective genes are expressed [150]. Furthermore, it has been shown that multiple *cis*-regulatory modules can compete with each other while present in the same TAD or regulatory hub, and that these regulatory hubs are made before there is an activation of transcription [151]. Hence, it appears that the formation of TADs and gene activation do not always need to be sequential.

Adding to this complexity, is the fact that enhancer–promoter distances and transcriptional states do not follow consistent correlations. While some studies have shown that with a decreasing enhancer–promoter proximity or the formation of loops by force there is an increase in gene transcription [79,152,153,154], other groups have shown that active transcriptional states correlate with an increase in enhancer–promoter distances [77,155]. Furthermore, regulatory analyses have shown that that multiple seemingly redundant or shadow enhancers work together to fine tune the robustness of gene expression [35,156,157]. Recently, it has also been shown that shadow enhancers can help to reduce the transcriptional noise that arises due to the fluctuating levels of TFs [158]. This raises a question as to how shadow enhancers and their counterparts coordinate interactions with promoters to modulate the activity and robustness of gene expression.

In eukaryotes, gene transcription has been observed to take place in bursts [153,159,160,161]. In the regulation of transcriptional bursting, it has been suggested that the events occur within condensates that contain all of the required regulatory components [40,42,43,45,162]. It has been proposed that the activation of transcription is a two-step process [39]. Initially, enhancers have to attain a “ready-to-go state”, evidenced by the presence of paused Pol II and coupled with being in close proximity to target promoters [150,163]. Secondly, in response to a cue, topological and spatial changes occur, leading to the activation of transcription [39]. If transcriptional activation occurs within the boundary of a condensate, then the mediator complexes, pre-initiation complexes and Pol II present within the condensate can act as bridges or connectors between enhancers and promoters [164]. In this model, strict physical enhancer–promoter contacts are not required, and a proximity range of 100–300 nm between enhancers and promoters may be sufficient to activate transcription [165]. To enhance the understanding of enhancer–promoter dynamics, high resolution imaging and, preferably, live imaging of transcriptional events alongside the visualization of allelic locations is required. This will give measurements that can address how enhancers dynamically interact either locally or globally and whether they do so on a single promoter one at a time or simultaneously on multiple promoters. These approaches can also give insight into how transcriptional activation corelates with changes in the chromatin structure. With respect to the *Hox* clusters, these approaches will be relevant for understanding how shared enhancers can coordinately regulate multiple genes and why some enhancers have selective preferences for some genes but not others.

### 2.4. Deciphering Enhancer Activity through Nascent Expression

It has been observed that enhancer contacts with *Hox* promoters can be made when enhancers are active but are also established irrespective of enhancer activity [166,167,168]. Enhancer contacts have been mapped through chromatin capture techniques, but their resolution has not been good enough to infer contacts with elements that are within a few kilobases of each other, which is the case in the vertebrate *Hox* clusters. However, there have been new techniques (Hi-M and Hi-CO) and improved Hi-C protocols that show a better resolution for inferring genomic contacts [169,170,171,172]. These techniques will aid in identifying and characterizing the proximal enhancer contacts that are relevant to transcriptional regulation. One way to map enhancer activity is to measure the transcriptional readout. Enhancer RNAs provide a means to infer active or inactive states of enhancers [74], while newly synthesized RNA or nascent transcripts of target genes can serve as a means of observing the transcriptional activation of genes.

Toward this goal, the development of several imaging and high throughput sequencing techniques has made it possible to identify nascent transcripts. Imaging methods are able to detect nascent transcripts via a single molecule fluorescent in situ hybridization (smFISH) [173,174], a Hybridization Chain Reaction (HCR) [175] and in a time lapse through the MS2/MCP stem loop system [176,177]. By combining MS2/MCP to monitor nascent transcripts and llama antibody tags to detect the corresponding proteins, it is possible to correlate the dynamics of rates of transcription with protein expression [178,179]. While these approaches have been extremely useful for insight into *Drosophila* development, to date it has been challenging to scale up these imaging methods beyond cell culture in different species in a high through-put manner. To infer the sequence of transcriptional activation by enhancers and bursting activities, it will be important to expand this to more diverse developmental contexts and species. There is a need to obtain live visualizations of the dynamics of nascent transcripts with cellular resolution, have a record of transcriptional activity from both alleles and to monitor the relative positions of enhancers regulating these patterns in the endogenous loci.

In genome wide analysis, with the advent of new sequencing technologies and computational programs, methods have been developed to detect or infer the presence of nascent transcripts. These methods employ the detection of RNA that is associated with chromatin, the enrichment of RNA that is associated with Pol II, or the detection of new RNA with Biotin or metabolic labelling [180]. These techniques enable the bulk detection of nascent transcripts in different tissues of interest. While each of these imaging and sequencing techniques are a step forward in the field, currently, each one has its own caveats and there is room to improve the detection of nascent transcripts in conjunction with genome organization. 

Progress has been made in the detection of transcriptional hubs containing a mediator complex using high resolution microscopy [43,164]. Recently, it has been observed that nascent transcripts themselves are in a feedback loop with the formation of transcriptional condensates; while nascent transcripts initially promote condensate formation, once they increase in numbers, they promote the dissolution of condensates [41,44]. This is relevant to the regulation of clustered *Hox* genes as it can be envisioned that enhancers and multiple gene promoters may reside within the same transcriptional hub or condensate for the coordinate regulation of their transcription. The gene promoters excluded from this hub would not be activated. Feedback from *Hoxb* nascent transcripts could, for example, promote transcription or lead to condensate dissolution (Figure 3B), renewing the cycle of transcriptional activation. Moreover, it has been shown that some TFs, through their activation domains, can form condensates with a mediator to activate genes [181] and some mediator subunits are known to interact with HOX proteins [182]. Further work needs to be undertaken to understand whether condensate formation and dissolution directly correlate with changes in enhancer contacts, and whether TFs such as HOX proteins have a direct role in affecting the enhancer contacts that result in gene activation or repression.

### 2.5. Additional Features in Regulation of Transcription of Hox Genes

Histone modifications along the *Hox* gene clusters and the 3D genome conformations within and around the clusters play a significant role in defining which *Hox* genes are going to be transcriptionally active [81,183,184,185]. This activation of *Hox* genes may be regulated by elements within the cluster, but it has also been seen that they can be regulated by sequences outside the cluster [35,167,186,187,188,189]. In addition, Polycomb Repressive Complexes (PRC1 and PRC2), which are involved in the silencing of *Hox* genes, effect chromatin compaction by post-translationally modifying histones [190] and, in turn, effect the transcriptional activation of *Hox* genes. Another layer of complexity for *Hox* genes is the presence of several non-coding RNAs within the gene clusters [191]. Non-coding RNAs can be widely expressed, display differential tissue expression patterns, have unique subcellular localizations or can participate in specific DNA, RNA and protein interactions to regulate chromatin [192]. Thus, the *Hox* cluster-specific non-coding RNAs may exhibit their own transcriptional signatures and inputs toward how *Hox* genes are transcriptionally activated [193].

In summary, exciting new ideas and insights are emerging on the regulatory mechanisms that govern the control of transcription. These offer new ways of thinking about how transcriptional events and processes are regulated in cell and developmental contexts. The *Hox* genes present a challenging and interesting system to understand the nuances of transcriptional control for gene activation [194]. The unique features of the *Hox* gene clusters (collinearity, high density of genes and regulatory elements, shifting TADs, etc.) and their important and conserved functional roles in development, disease and evolution provide an important paradigm for investigating transcriptional regulation in a deeper and more detailed mechanistic manner. Furthermore, from an evolutionary perspective, it will be interesting to explore whether conserved GRNs that involve *Hox* genes, such as the one for hindbrain segmentation [19,29,195,196], utilize the same or different transcriptional mechanisms in the regulatory circuits modulating conserved programs of expression. With the advent of new imaging and sequencing technologies, gaining insight into how enhancers function in the complex dynamics of the transcriptional regulation of *Hox* genes is beginning to enter an exciting new phase.

## Figures and Tables

**Figure 1 jdb-10-00004-f001:**
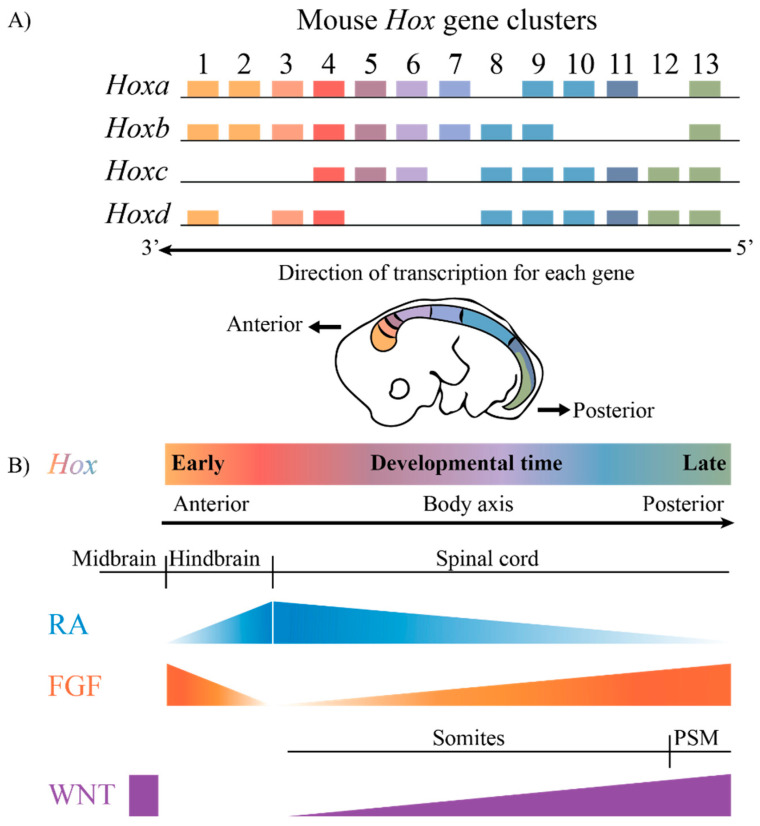
The mammalian *Hox* gene clusters and the conserved signaling pathways that play a role in defining the *Hox* gene expression profiles. (**A**) In mammals, there are four clusters of *Hox* genes, each on different chromosomes. They exhibit spatial and temporal collinearity, such that 3′ *Hox* genes are expressed early in development as well as more anteriorly in an embryo generating nested domains of expression as depicted in the drawing of an E10 mouse embryo. (**B**) The restricted domains of *Hox* expression arise through an integration of signaling molecules such as RA, FGF and WNT, which are expressed in gradients along the embryonic axis. PSM, presomitic mesoderm.

**Figure 2 jdb-10-00004-f002:**
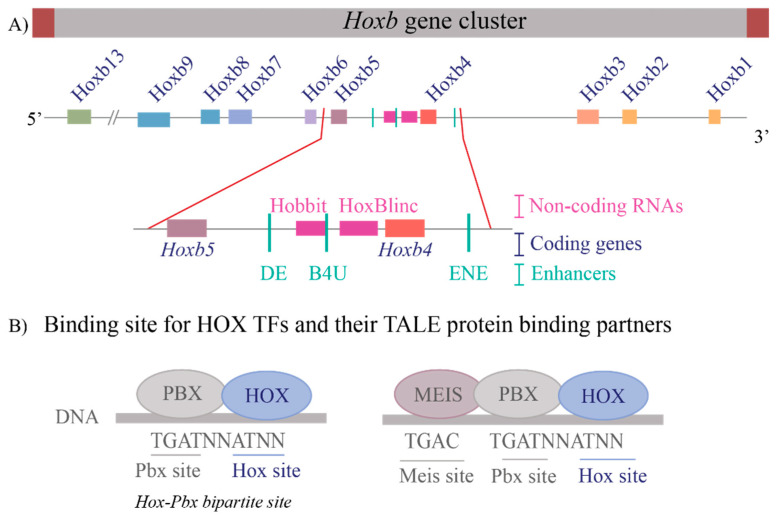
Transcriptional complexity of the *Hoxb* gene cluster and binding of HOX Transcription factors to DNA (**A**) A drawing of the *Hoxb* gene cluster to illustrate that non-coding RNAs as well as enhancers that contain RAREs (Retinoic Acid Response Elements) are interspersed within the coding *Hox* genes. The enlargement of the *Hoxb4-Hoxb5* region shows the complexity within the region that contains three RAREs, two present upstream of *Hoxb4* and one present downstream of *Hoxb4* and two non-coding RNAs, *Hobbit* and *HoxBlinc*. Brown boxes flank the cluster depict boundary elements, colored squares are different *Hox* genes, pink boxes are non-coding RNAs, and green lines represent *RARE* enhancers. (**B**) Depicts the consensus DNA binding sites for HOX proteins and their binding partners, the TALE proteins PBX and MEIS. HOX proteins can bind on *Hox-Pbx* bipartite sites, or they can bind on DNA in ternary complexes along with both PBX and MEIS. Blue ovals are HOX proteins, and grey ovals are TALE protein binding partners.

**Figure 3 jdb-10-00004-f003:**
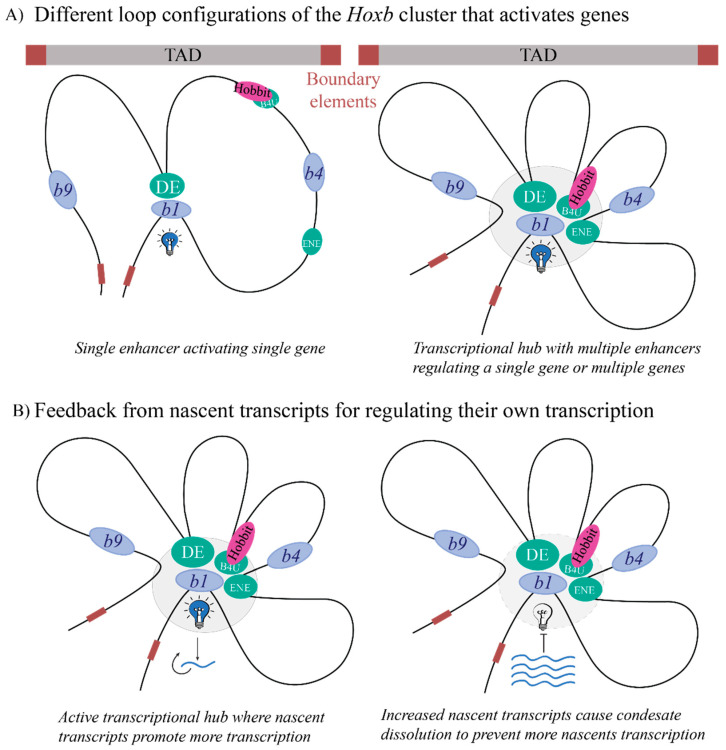
Schematic of how the chromatin may loop to activate genes within the transcriptionally complex *Hoxb* gene cluster (**A**) Different loop confirmations envisioned for activation of specific genes within the *Hoxb* cluster, which contains several enhancer elements (such as DE, B4U and ENE) as well as coding and non-coding genes. (**B**) Inference [41,44] for how nascent *Hoxb* transcripts may promote condensate formation to increase nascent transcription, and subsequently, how an increased number of nascent transcripts may inhibit transcription by promoting condensate dissolution. Brown boxes at cluster edges depict boundary elements, blue and pink colored ovals depict coding genes and non-coding RNAs, respectively, and green ovals are RARE enhancers.

## Data Availability

All data and results referred to in this review have been previously published and cited with the appropriate reference.

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
