# Peer review of "Transcriptional Regulation and Implications for Controlling Hox Gene Expression"

_jdb, 2022, doi:10.3390/jdb10010004_

Round 1

Reviewer 1 Report

Dear authors,

I really enjoyed your review. It really focuses on that area of transcription regulation science which is actively developing thanks to new methods. Now is a good time to use new data to build new theories. I was particularly pleased with your conscientious approach to selecting references to modern research that give readers a comprehensive overview of the topic.  I have one suggestion and one comment.

  1. Since your review mainly focuses on those mechanisms of regulation that involve the work of enhancers, perhaps this should be reflected immediately in the title.
  2. Figure 3(A) shows two identical TADs as grey rectangles. From the caption to the illustration and from the illustration itself, it is not clear why they are shown. Maybe the boundary elements should be marked on the loops themselves?

Author Response

We have made changes to address the reviews suggestions. We could not find an easy way to include the word enhancers into the title, but we have added “enhancer” to the key words. We also modified Figure 3 to include the boundary elements in the drawing of loops as suggested by the reviewer. The other changes are primarily in the Figure legends to increase clarity in understanding the information in the figures. 

Reviewer 2 Report

The manuscript is well written and really interesting. In particular, the contribution given by Dr. Krumlauf's experience in the study of HOX genes can be understood reading this review. Understanding the mechanism of action, carried out by this network has strong repercussions in the study of diseases related to the deregulation of one or more HOX genes. In fact, it has already been demonstrated that HOX genes, like transcription factors, are related to the control of the cellular memory program, a biological process responsible to control the cell phenotype. In light of this, the deregulation of HOX cluster determines morphological and/or physiological disease. In conclusion, the authors have written a review in order to better understand some aspects related to the transcription process that help to clarify how HOX network works and which is the transcriptional mechanism. Therefore, I consider the review available for publication.

Author Response

Thank you for the positive comments.